# Determination of Triacylglycerol Composition in Mealworm Oil (*Tenebrio molitor*) via Electrospray Ionization Tandem Mass Spectrometry with Multiple Neutral Loss Scans

**DOI:** 10.3390/insects15050365

**Published:** 2024-05-17

**Authors:** Seongeung Lee, Minkyoung Kim, Hyeokjun Cho, Gyeong-Hwen Lee

**Affiliations:** Lotte R&D Center, 201, Magokjungang-ro, Gangseo-gu, Seoul 07594, Republic of Korea; minkyoung.kim@lotte.net (M.K.); hyeokjun_cho@lotte.net (H.C.); ghlee1@lotte.net (G.-H.L.)

**Keywords:** lipidomics, insect lipid, fatty acid, triacylglycerol, shotgun analysis

## Abstract

**Simple Summary:**

Mealworms are edible insects that are used as food ingredients because nearly all of their edible sections are lipid and protein sources that humans can consume. Triacylglycerols are significant energy sources and the main components of mealworm oil. Due to the composition of triacylglycerol being related in basic properties to oils, the determination of the triacylglycerol composition is important to understand oils. Our study used electrospray ionization tandem mass spectrometry with multiple neutral loss scans to analyze the content and composition of triacylglycerol in mealworm oil. The acquired scan data were deconvoluted to minimize isotopic interference. Additionally, the validation parameters showed relatively good results. The mealworm oil mainly contained triacylglycerol possessing palmitic acid, oleic acid and linoleic acid. The influence of triacylglycerol on mealworm oil is relatively stronger against oxidative stress compared to other vegetable oils.

**Abstract:**

Mealworms (*Tenebrio molitor*) have been used as an alternative source of proteins and lipids. Triacylglycerols (TAGs) are major sources of energy and have been used to provide essential fatty acids. They are also the main components of mealworm oil, and their composition and content are extensively linked to its physical and chemical properties. However, because of the complexity of TAG molecules, their identification and quantitation are challenging. This study employed electrospray ionization tandem mass spectrometry (ESI-MS/MS) with multiple neutral loss scans (NLS) to analyze the TAG composition and content in mealworm oil. Identifying and quantifying TAGs using ESI-MS/MS in combination with multiple NLS was an efficient way to improve accuracy and timeliness. For the accurate quantification of TAGs, isotopic deconvolution and correlation factors were applied. A total of 57 TAGs were identified and quantified: C52:2 (16:0/18:1/18:1) (1549.4 nmol/g, 18.20%), C52:3 (16:0/18:1/18:2) (1488.1 nmol/g, 17.48%), C54:4 (18:1/18:1/18:2) (870.1 nmol/g, 10.23%), C54:6 (18:1/18:2/18:2) (659.8 nmol/g, 7.76%) and C52:4 (16:0/18:2/18:2) (600.5 nmol/g, 7.06%), which were the most abundant TAGs present in the mealworm oil. The fundamental properties of mealworm oil, including its degree of oxidation, nutritional effect and physical properties, were elucidated.

## 1. Introduction

Edible insects have been consumed in many countries, including Africa, East Asia and Southeast Asia, for approximately 5000 years [1]. Sustainable and green foods have become increasingly important worldwide as substitutes for meat and meat products. Edible insects can be a great solution to satisfy this demand because they show a higher feed conversion efficiency than other domestic animals and emit comparatively lower amounts of greenhouse gases, including carbon dioxide, ammonia and methane [2]. Additionally, insects have been known to contain valuable nutrients, including lipids, proteins, minerals and vitamins [3,4,5]. Among these nutrients, protein is the most abundant ingredient, followed by lipids [6].

The mealworm (*Tenebrio molitor*), an edible insect, is utilized as a food source and animal feed. It is also recognized as an alternative source of lipids and proteins for livestock. This is because they contain edible protein and lipid sources for humans, and the percentage of edible parts is over 90% [7]. Additionally, the value of mealworm nutrients is higher than plant nutrients in terms of proteins, vitamins and minerals [8]. According to Alves et al. [9], they contain an abundant lipid content, and similarly with proteins. They mainly comprise abundant polyunsaturated fatty acids, including linoleic and linolenic acids. [10].

Lipids, including triacylglycerols (TAG), phospholipids, sterols and fat-soluble vitamins, play important roles in the human body. They are major sources of energy, providing essential fatty acids and have utility as antioxidants [11]. Previous studies have shown that lipids in insects have many bioactive compounds, such as phospholipids, fat-soluble vitamins, sterols and carotenoids [12,13,14]. TAG is the most abundant lipid in edible oils and fats, including palm, olive and lard oils. According to Mateos et al. [15], the composition and content of TAG are closely related to the lipid oxidative stability. In their study, the oxidative stability was evaluated by an accelerated automated test using a Rancimat apparatus (model CH 9100, Metrohm Co., Basel, Switzerland). TAG contains linoleic acid which shows a relatively higher oxidation rate, while on the other hand, TAG contains saturated fatty acids, such as palmitic acid and stearic acid, which show a relatively low oxidation rate. Also, the TAG composition and its regioisomeric properties are mainly related to the physical characteristics of lipids and fats [16,17]. For example, even though TAG is composed of the same fatty acids, including 18:0, 18:1 and 18:2, the melting temperature of 18:0/18:2/18:1 and 18:0/18:1/18:2 showed a difference of about 10 °C [18]. Therefore, analyzing the composition, content and regioisomeric properties of TAG species is important for a more specific understanding of the general properties of mealworm oil. However, an analysis of the TAG composition is complicated because it has a complex molecular structure. TAG consists of one glycerol molecule with three fatty acid chains. In particular, fatty acids have a variety of characteristic features influenced by carbon numbers and degree of unsaturation [19].

Various analytical methods have been reported for the determination of TAG compositions. For example, Han and Ye [20] and Wang et al. [21] reported an analytical method for TAGs using thin-layer chromatography (TLC). Toschi et al. [22] and Buchgraber et al. [23] reported an analytical method using gas chromatography with a flame ionization detector (GC-FID) and liquid chromatography with an evaporative light scattering detector (LC-ELSD), respectively. Beccaria et al. [24] and Cheong et al. [25] reported an analytical method using liquid chromatography with tandem mass spectrometry (LC-MS/MS). In particular, Gowda et al. [26] reported an untargeted lipidomic analytic method using HR-MS to determine lipid compounds, such as TAG, in mealworms. Among these methods, mass spectrometry has various scan modes, including neutral loss scans (NLS), multiple reaction monitoring (MRM) and selective ion monitoring (SIM). Recently, multiple NLS have been used to determine TAG compositions and contents. The NLS continuously determines the mass-offset scanned data between the first and second analyzers [27]. In other words, the first analyzer scans the chemical components in the samples and then the second analyzer scans fragment ions that react in the collision cells. In this process, the mass offset between the two analyzers is computed, leading to the detection of neutral loss molecules with a mass of the target molecules. It is an effective scan method for identifying lipids that consist of fatty acyl chains.

Direct infusion shotgun analysis with electrospray ionization (ESI) is an effective technique, especially for the high-throughput analysis of lipids [28]. It provides direct MS scans for multiple NLS and is very flexible and fast when compared with LC-MS/MS. Han et al. [29] determined TAG in mouse organs, such as the heart, muscle and liver, using a shotgun lipidomic analytical method. Several studies have analyzed the TAG compositions of arabidopsis seeds, salmon tissue and cold rapeseed oil using direct infusion methods with multiple NLS [30,31,32]. However, this method has not yet been applied to study the TAG content and its composition in mealworm oil. Moreover, other studies reported the composition of minerals, proteins and lipids, including major fatty acids, in mealworms [10,26,33,34,35,36,37,38]. But, studies on TAGs are limited.

Therefore, the objective of this study was to analyze the composition and content of TAGs in mealworm oil using ESI-MS/MS with multiple NLS. The multiple NLS data obtained were deconvoluted to reduce isotopic interference. These findings can be utilized to determine the general properties of the mealworm oil.

## 2. Materials and Methods

### 2.1. Materials

Mealworm oil produced in Gyeongsangbuk Province, Korea, was purchased online. To analyze the fatty acid composition, 14% boron trifluoride in a methanol solution, a 37 component fatty acid methyl ester (FAME) standard mixture and a sodium hydroxide solution were purchased from Sigma Aldrich Chemical Co. (St. Louis, MO, USA). For the analysis of the TAG composition, the standards included triundecanoin (C33:0, tri-11:0, the internal standard for fatty acid analysis), tridodecanoin (C36:0, tri-12:0), trimyristin (C42:0, tri-14:0), tripalmitolein (C48:3, tri-16:1), tripalmitin (C48:0, tri-16:0), triheptadecanoin (C51:0, tri-17:0), tristearin (C54:0, tri-18:0), triolein (C54:3, tri-18:1), trilinolein (C54:6, tri-18:2), trilinolenin (C54:9, tri-18:3), trinonadecanoin (C57:0, tri-19:0), tripentadecanoin (C45:0, tri-15:0), triheptadecenoin (C51:3, tri-17:1, the internal standard for TAG), trinonadecenoin (C57:3, tri-19:1), trinonadecadienoin (C57:6, tri-19:2), trieicosanoin (C60:0, tri-20:0), trieicosenoin (C60:3, tri-20:1), trieicosadienoin (C60:6, tri-20:2) and trieicosatrienoin (C60:9, tri-20:3) were purchased from Nu-check Prep, Inc. (Elysian, MN, USA). Other solvents, such as methanol, chloroform and n-hexane, were purchased from J.T. Baker (Phillipsburg, NJ, USA).

### 2.2. Analysis of Fatty Acid

For analyzing fatty acids, a modified version of the method of the Korea Food Code [39] was followed. Briefly, mealworm oil samples (50 mg) were mixed with 0.5 N sodium hydroxide in a methanol solution and a 1 mL internal standard (glyceryl triundecanolate, 1.0 mg/mL in hexane) in a glass tube (22 mm × 150 mm). The mixed sample was heated for 5 min at 100 °C. In this process, the fatty acid in the glyceride has been changed to alkaline salts of fatty acids. Subsequently, it was incubated for 30 min at 100 °C with 14% borontrifluoride in a methanol solution. In this process, the alkaline salts of the fatty acids were converted into FAME. After heating, 1 mL of n-hexane and 5 mL of a saturated sodium chloride solution were added, mixed and centrifuged. The upper layer was filtered and injected into the GC-FID (Model 7890a, Agilent, Santa Clara, CA, USA) with a fused-silica capillary column coated with 100% biscyanopropyl polysiloxane as the stationary phase (SP-2560, 100 m × 0.25 mm with a 0.2 µm thickness, Supelco, Bellefonte, PA, USA) to analyze FAME. The inlet condition was a split ratio of 200:1 at 250 °C. Nitrogen was used as a carrier gas at a flow rate of 1 mL/min. The initial temperature of the oven was 100 °C for 3 min, which was then ramped to 190 °C for 5 min, followed by another ramping to 240 °C at a rate of 3 °C/min and, finally, kept at that temperature for 25 min. FAMEs were quantified based on their peak areas using an internal standard and conversion coefficient. Fatty acid analysis is an important step in multiple NLS because the fatty acid composition and molecular weight provide fundamental information.

### 2.3. Analysis of Triacylglycerol

The analysis of the TAGs followed the method of Lee et al. [40] with some modifications. Briefly, mealworm oil samples (100 mg) were added to a glass tube and dissolved in 10 mL of a solvent (chloroform:methanol = 2:1). The dissolved oil sample was vortexed for 10 min, to which 0.5 mL of the internal standard (C51:3, tri-17:1, 0.5 nmol) was added. Then, 4 mL of ammonium acetate solution (chloroform:methanol:300 mM, ammonium acetate = 30:67:3) was added, and the mixture was vortexed for 2 min. In this process, the TAGs were converted to ammonium-adducted TAG through the reaction of ammonium acetate. The TAGs were analyzed using direct infused ESI-MS/MS (Model Xevo TQ-S, Waters, Milford, MA, USA) in positive mode. The sample was infused into an ion source at a flow rate of 20 µL/min. The ion source temperature was 450 °C with a 4.0 kV capillary voltage and 60 V cone voltage. The gas flow rates for the desolvation and cone were 600 and 150 L/h, respectively. Nitrogen was used as the collision gas, with a collision energy of 30 V. The scans targeted the loss of fatty acyl chains as neutral fragments with ammonium ions, including NLS 245.2 (14:0), NLS 271.2 (16:1), NLS 273.2 (16:0), NLS 285.2 (17:1), NLS 295.2 (18:3), NLS 297.2 (18:2), NLS 299.2 (18:1), NLS 301.2 (18:0), NLS 327.2 (20:1) and NLS 329.2 (20:0). The acquired data were processed using the MassLynx software, version 4.1 (Waters, Milford, MA, USA). The intensity of the obtained spectrum was deconvoluted to reduce isotopic interference and improve quantification accuracy. The calculation of isotopic deconvolution followed Li et al. [19], and the detailed process is explained in the Section 3. To quantify the TAG, the corrected spectrum intensity was applied to statistically estimated equations using adjustment factors. The adjustment factor was calculated using the correlation between the commercial TAG and the internal standard [30]. Factors which were unavailable for calculation were estimated using a regression curve generated by statistical software SPSS, version 25 (IBM, Armonk, NY, USA).

### 2.4. Method Validation

The method validation followed the method of the IUPAC technical report [41]. The parameters of validation included trueness (recovery), precision (repeatability), linearity, limit of detection (LOD) and limit of quantitation (LOQ). Recovery was obtained as the percentage difference between the quantity of TAG recovered from the spiked and non-spiked samples divided by the quantity of TAG added (10, 50 and 100 μM spiked). The precision of the assay was presented as the coefficient of variation of at least five repetitions. The linearity, LOD and LOQ were checked using eight TAG standards, including tripalmitin, tristearin, triolein, trilinolein, trilinolenin, 1,3-dipalmitoyl-2-oleoylglycerol, 1,3-distearoyl-2-oleoylglycerol and 1-palmitoyl-2-oleoyl-3-stearoyl-sn-glycerol at concentrations in the range of 0.5–100 µM for multiple NLS.

## 3. Results

### 3.1. Fatty Acid in Mealworm Oil

The fatty acid composition and content in mealworm oil was determined using GC-FID. Fatty acids, including myristic (C14:0), palmitic (C16:0), palmitoleic (C16:1), stearic (C18:0), oleic (C18:1), linoleic (C18:2), linolenic (C18:3), arachidic (C20:0) and eicosenoic acids (C20:1) were detected (Figure 1). The fatty acid contents and compositions in the mealworm oil are shown in Table 1. In our study, mealworm oil showed the highest oleic acid content among the fatty acids (48.68 g/100 g). The content of linoleic acid, a major polyunsaturated fatty acid that produces the lipid components of all cell membranes in the body, was high (23.09 g/100 g). Palmitic acid, a well-known saturated fatty acid, also showed a relatively high content in mealworm oil (17.75 g/100 g). Myristic, palmitoleic, stearic, linolenic, arachidic and eicosanoic acids were less abundant, occurring at levels below 5.0 g/100 g.

### 3.2. Triacylglycerol in Mealworm Oil

Multiple NLS is an effective scanning method for the identification of TAGs that have a fatty acyl chain, such as a neutral loss structure. Multiple NLS conditions were established using the fatty acid composition determined by GC-FID. The range of multiple NLS was m/z 700–1200. The spectra obtained from multiple NLS were used to identify the combination of fatty acyl chains and to confirm each TAG using the molecular mass of the ammoniated ion form. For instance, the spectrum at 897.2 was found in NLS 295.2, 297.2 and 299.2, indicating the loss of C18:1, C18:2 and C18:3 fatty acids from its TAG compounds (Figure 2). In addition to the identification of the three fatty acyl chains determined using multiple NLS, the masses of the ammoniated ions of TAG matched 897.2, leading to the confirmation of C54:6 as 18:1/18:2/18:3 or 18:2/18:2/18:2. This process was repeated using different NLS conditions for fatty acids to identify TAG in mealworm oil. A total of 57 TAGs were detected in the range of m/z 793.0–933.0 in NLS in positive mode. Most of the TAGs identified in the multiple NLS were composed of palmitic, oleic and linoleic acids, similar to the fatty acid analysis data.

The spectra of the identified TAGs were deconvoluted to reduce isotopic interference. The main atoms of TAG are carbon, oxygen and hydrogen, which have their own isotopes that cause interference in the mass spectrometry determination. In particular, this interference is influenced by carbon variants such as ^13^C and ^12^C. Therefore, one TAG molecule (M) may have isotopes such as M + 1, M + 2 and M + 3, and some of them may interfere with different masses [19]. Isotopic deconvolution was performed, as described by Li et al. [19]. First, Z_M_ was calculated using the following formula:(1)$$Z_{M}=\left[\left(N_{C}\times 1.12\right)\times \left(N_{C}\times 1.12\right)/200+\left(N_{O}\times 0.204\right)\right]/100$$
where N_C_ and N_O_ are the number of carbon and oxygen atoms in the diacylglycerol (DAG) product ion from the fragmented TAGs, respectively. Subsequently, Z_M_ was utilized in the following formula to compute the corrected peak intensity:(2)IM+2  C=IM+2−IM  C×ZM
where I_M + 2_ is the raw spectrum intensity and ^C^I_M_ is the corrected spectrum intensity of molecule M. In this process, 58 TAG spectra were deconvoluted depending on the sample, and the C50:0 (16:0/16:0/18:0) TAG spectrum was removed because of isotopic interference. The partial results of the isotopic deconvolution in NLS 299.2 are shown in Figure 3. The spectrum at m/z at 897.2 showed the same intensity after isotopic deconvolution. On the contrary, the spectrum at m/z at 907.2 showed the greatest decrease in intensity after isotopic deconvolution. This spectrum was removed owing to isotopic interference.

After isotopic deconvolution of the spectrum signal obtained for individual NLS, the correlation factors were used to correct the variable NLS spectrum of the acquired TAGs that varied in the number of carbon and double bonds in the fatty acyl chain. The factor is defined as a coefficient, where the NLS spectrum signal of the internal standard (C51:3) is divided by the NLS spectrum signal of each TAG when the TAG and internal standard have the same concentration [30]. The factor was calculated using commercially purchased TAG standards, and a statistical estimation for the regression curve was conducted using the calculated factors. First, a regression curve for saturated TAG was obtained using the calculated adjustment factor for the saturated TAGs (Appendix A). Subsequently, regression curves were obtained for the unsaturated TAG, including three double bonds (Appendix A) and six double bonds (Appendix A). Finally, the final regression curve was deduced for each carbon of the TAGs (46–56) using the factor acquired from the calculation and estimation (Figure 4). These factors were used for the deconvoluted spectral intensity of the NLS. For instance, the correlation factor for m/z at 877.2 (C52:2) was 0.91, which indicates that 0.91 moles of C52:2 and 1.0 mole of the internal standard (C51:3) would show the same spectrum intensity. Thus, the m/z at 877.2 (C52:2) from NLS 273.2 was corrected by multiplying it by 0.91. The increase in the factor indicated a decrease in the intensity of the NLS spectrum, which increased with the number of carbons in the range of C46–C60 in the fatty acid chain. These results are similar to those of Li et al. [30] and Lee et al. [38], in which TAGs with short fatty acid chains were shown to have higher sensitivity than those with long fatty acid chains. The adjustment factor increased depending on the increase in the unsaturation of TAGs in the range of 3–6 double bonds. However, it decreased in the range of 0–3 double bonds in the TAGs, which indicated the highest intensity of the NLS spectrum for the three double bonds. Han et al. [27] reported a similar tendency in their study.

The contents of individual TAG species in mealworm oil were quantified using a spectrum acquired from multiple NLS with isotopic deconvolution and correlation factors. The TAG quantification was performed following Li et al. [30]. First, individual fatty acids in TAGs (A_m_) at a particular m/z were computed (in nmol) using the following formula:(3)$$Am=\frac{\left(\mathrm{observed}\;\mathrm{TAG}\;\mathrm{intersity}\times \mathrm{internal}\;\mathrm{standard}\;\mathrm{amount}\right)\times \mathrm{adjustment}\;\mathrm{factor}}{\mathrm{observed}\;\mathrm{internal}\;\mathrm{standard}\;\mathrm{intensity}}$$

The computed amount of fatty acids in the TAG at each m/z value was used to calculate the amount of each TAG molecule. For instance, the TAG mass m/z at 875.0 (C52:3) contained various fatty acid chains. The highest number of fatty acid chains at C52:3 was C16:0, C18:1 and C18:2. Based on the number of carbons and double bonds in the three fatty acid chains, three types of fatty acid combinations were deduced for C52:3, including 16:0/18:1/18:2, 16:1/18:1/18:1, and 16:1/18:0/18:2. The amounts of 16:0/18:1/18:2 were represented by the amounts of 16:0, A_16:0_. The 16:1/18:1/18:1 have two C18:1 acyl chains. Thus, the amount of TAG was calculated using the formula A_18:1_/2. In addition, the levels of 16:1/18:0/18:2 were represented by the amounts of C18:0 and A_18:0_.

Individual TAGs (57) in mealworm oil identified by their fatty acyl chains were computed, and the results are shown in Table 2; the results are presented as nmol/g and % portion, respectively. The major TAGs in mealworm oil were C52:2 (16:0/18:1/18:1) (1549.4 nmol/g, 18.20%), C52:3 (16:0/18:1/18:2) (1488.1 nmol/g, 17.48%), C54:4 (18:1/18:1/18:2) (870.1 nmol/g, 10.23%), C54:5 (18:1/18:2/18:2) (659.8 nmol/g, 7.76%) and C52:4 (16:0/18:2/18:2) (600.5 nmol/g, 7.06%), accounting for 60.73% of TAGs in mealworm oil. TAGs in mealworm oil had a high frequency of palmitic, oleic and linoleic acids; in particular, 77.51% of the identified TAGs possessed at least one oleic acid as a fatty acyl chain in their molecule. Palmitic, linoleic and linolenic acids also showed a high frequency (5.85–30.73%).

### 3.3. Method Validation

The TAG compositional analysis method was validated by determining the trueness (recovery), precision (repeatability), linearity, LOD and LOQ. The trueness was evaluated by analyzing mealworm oil to 10, 50 and 100 µM of the TAG standards, including tripalmitin, tristearin, triolein, trilinolein, trilinolenin 1,3-dipalmitoyl-2-oleoylglycerol, 1,3-distearoyl-2-oleoylglycerol and 1-palmitoyl-2-oleoyl-3-stearoyl-sn-glycerol, were added prior to extraction. The percent mean recovery ± standard deviation (*n* = 5) was 90.9 ± 3.2, 86.9 ± 2.4, 95.0 ± 0.9, 93.6 ± 4.5, 94.2 ± 2.8, 88.1 ± 3.6, 90.6 ± 1.6 and 92.4 ± 1.9 for eight separate TAGs (Table 3). The repeatability (relative standard deviation, RSD) was usually less than 11.0%, except for C50:3 (14:0/18:0/18:3) and C52:4 (16:1/18:1/18:2). The linearity was checked using the same TAG standard at concentrations in the range of 0.5–100 µM for multiple NLS. All the standard curves were linear, with a correlation coefficient of 0.999 (Table 3). The range of LOD was from 0.05 µM to 0.25 µM and the LOQ was from 0.1 µM to 0.5 µM, depending on the TAG. The values of the validation parameters were relatively high.

## 4. Discussion

In our results, mealworm oil showed relatively higher contents of palmitic acid, oleic acid and linoleic acid compared to other fatty acids, including myristic, palmitoleic, stearic, linolenic, arachidic and eicosanoic acids. The most abundant fatty acid in mealworm oil was oleic acid, which was also reported as the highest content of fatty acid in other studies [9,37,38]. According to Alves et al. [9], the second most abundant fatty acid in mealworm oil cultivated in Brazil was palmitic acid, while on the other hand, the second most abundant fatty acid in mealworm oil in other studies, including our research, was linoleic acid [37,38]. In particular, fatty acid composition data cultivated in Korea showed considerable similarity to our results where the fourth and fifth most abundant fatty acids were myristic acid and stearic acid, respectively [38]. The fatty acid contents and compositions in mealworm oil were different, depending on parameters such as feed type, raising conditions and physiological characteristics [42,43]. According to Francardi et al. [43], a significant increase in linolenic acid was shown in mealworm oil when feed enriched with linseed oil was provided, indicating a strong association between the fatty acid profile of mealworm oil and diet. Essential fatty acids have been mainly found in marine species, such as mackerel, demonstrating that they can be used for various purposes, including food supplements for humans, feed for livestock and recycling supplements. Among these fatty acids, linolenic acid is a biofunctional lipid; it has been utilized as a precursor for the biosynthesis of healthy polyunsaturated fatty acid, including docosahexaenoic acid (DHA) and eicosapentaenoic acid (EPA). Although mealworms have abundant linolenic acid that is recognized as trace component for nutrition in insects, DHA and EPA was not detected [44]. Mealworm oil contained a relatively higher amount of unsaturated fatty acids, similar to vegetable oils. Unsaturated fatty acids, including oleic, linoleic and linolenic acid, are responsible for lowering blood pressure and cholesterol levels in human blood [45]. However, the abundant amount of unsaturated fatty acid in mealworm oil makes it vulnerable to oxidative stress [46]. Thus, the storage conditions for mealworm oil are important for reducing oxidative damage. The fatty acid composition from this study and the molecular weight of the ammoniated ions were used based on the conditions of multiple NLS.

TAGs in mealworm oil had a high abundance of palmitic acid (16:0), oleic acid (18:1) linoleic acid (18:2), in particular, approximately 53.0% of identified TAGs possessed at least one oleic acid as an aliphatic chain in their structure. The major TAG species in mealworm oil were C52:2 (16:0/18:1/18:1), C52:3 (16:0/18:1/18:2), C54:4 (18:1/18:1/18:2), C54:5 (18:1/18:2/18:2) and C52:4 (16:0/18:2/18:2). In previous studies, vegetable oils such as rapeseed, grapeseed, sunflower and soybean oil also contained a high abundance of oleic acid and linoleic acid. Thus, their major TAG species were C54:3 (18:1/18:1/18:1), C54:4 (18:1/18:1/18:2) and C54:6 (18:2/18:2/18:2) [29,47,48]. They contained relatively similar TAG species, however, mealworm oil contained a relatively high abundance of the TAGs possessing palmitic acid in their structure. According to Mateos et al. [15], a TAG that contains at least one palmitic acid showed a relatively low oxidation rate. Thus, the mealworm oil is relatively stronger against oxidative stress than other vegetable oils.

Tzompa-Sosa et al. [37] determined the equivalent carbon number (ECN) of mealworms using GC-FID. The ECN is determined by the formula ECN = N − 2n, in which N is the number of carbon atoms in the three FA that make up the TAG and ‘n’ is the total number of double bonds present in the TAG. In their results, mealworms showed high concentrations of glycerides with an ECN of 50–54 and low concentrations of glycerides with an ECN of 36–38. However, if we calculate the ECN number based on our results, it is estimated that 46 and 48 are the highest. This difference is thought to be due to differences in equipment, and there will be a need to study the differences between GC-FID and ESI-MS/MS in the future.

Gowda et al. [26] determined the TAG composition in yellow mealworms using untargeted LC-HR-MS. They indicated that TAGs with monounsaturated or polyunsaturated fatty acids, such as palmitic, oleic and linoleic acids, are the major molecular species in mealworms. The high amount of polyunsaturated fatty acid from TAGs means that mealworms are recognized as a high energy source. Therefore, the mealworm can be used for developing food and feed products. However, the TAG content was different from our results. In their study, TAG C50:4 was found to be the most abundant relative to its concentration. TAG C50:5 and C48:3 showed relatively high concentrations in the yellow mealworm samples. Variations in the TAG composition influenced the sample status of raw mealworms and mealworm oils. Moreover, cultivation parameters, such as feed type and composition, raising conditions and physiological characteristics, affected the variation in TAG composition.

The other analysis methods such as HPLC, TLC and GC-FID have been used to analyze the TAG compositions and contents in edible oils, such as palm, rapeseed and soybean oils, owing to their relatively accurate values [29,36]. However, quantitative literature on the absolute amounts of TAG species, for example, in nmol/g, is still lacking for mealworm oils. A quantitative TAG profiling method for mealworm oil based on multiple NLS is described in the present study. Although our study provided quantitative results for TAG in mealworm oil with information on which fatty acids were attached to the glycerol backbone, the sn-positions of the acyl chains were not specified in the description of the TAG. Therefore, further research on the positional fatty acid composition (sn-2 and sn-1,3) of TAG is needed.

## 5. Conclusions

ESI-MS/MS with multiple NLS was effectively used to determine the TAG composition and content in mealworm oil. Isotopic deconvolution and adjustment factors of various TAGs were used for precise quantification. A total of 57 TAGs were identified and quantified: C52:2 (16:0/18:1/18:1) (1549.4 nmol/g, 18.20%), C52:3 (16:0/18:1/18:2) (1488.1 nmol/g, 17.48%), C54:4 (18:1/18:1/18:2) (870.1 nmol/g, 10.23%), C54:6 (18:1/18:2/18:2) (659.8 nmol/g, 7.76%) and C52:4 (16:0/18:2/18:2) (600.5 nmol/g, 7.06%), being the most abundant TAGs. Using direct infusion LC-MS/MS with repeated NLS to identify and quantify TAG was an efficient approach for improving the detection accuracy. In addition, the validation parameters of this method showed relatively good results, such as the recovery which showed 90–100%, except for tristearin and 1,3-dipalmitoyl-2-oleoylglycerol. Also, all the standard curves were linear, with a correlation coefficient of 0.999. This approach effectively revealed the composition and content of TAG in mealworm oil. This has ramifications for the physical and chemical properties and can be used to understand phenomena such as oxidation levels and other properties. However, although our study provided quantitative results for TAG in mealworm oil with information on which fatty acids were attached to the glycerol backbone, the sn-positions of the acyl chains were not specified in the description of TAG. Therefore, further research on the positional fatty acid composition (sn-2 and sn-1,3) of TAG is needed.

## Figures and Tables

**Figure 1 insects-15-00365-f001:**
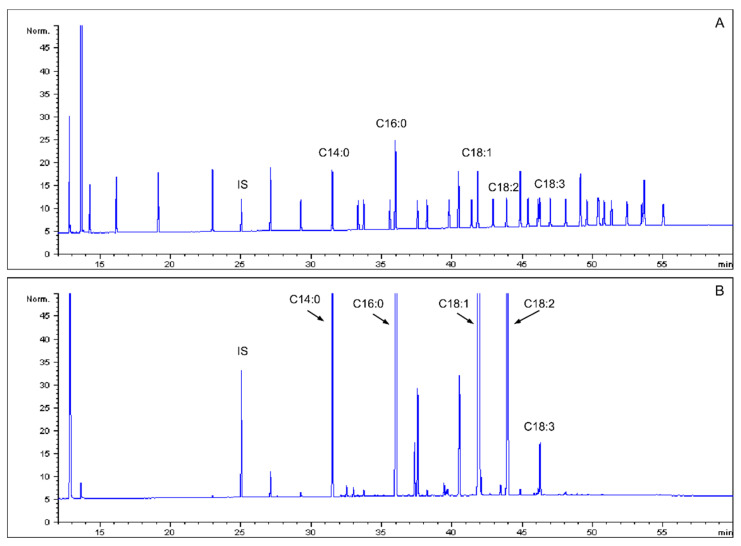
The chromatogram of FAME in mealworm oil. (**A**) FAME 37 standard. (**B**) The mealworm oil sample.

**Figure 2 insects-15-00365-f002:**
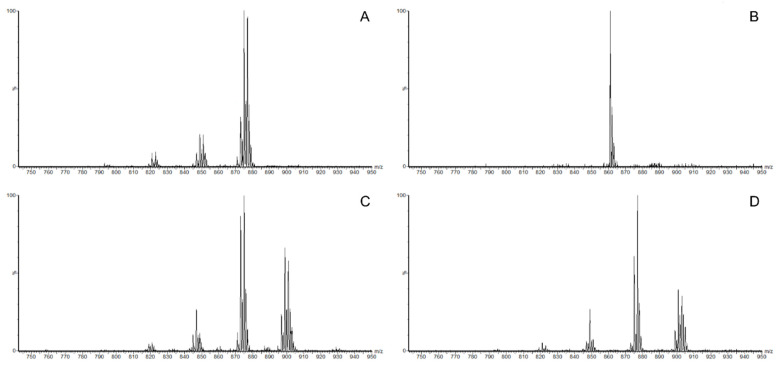
Partial spectra of multiple NLS of ammoniated TAG ions in mealworm oil. (**A**) NLS 273.2 (Loss of C16:0). (**B**) NLS 285.2 (Loss of C17:1, internal standard). (**C**) NLS 297.2 (Loss of C18:1) (**D**) NLS 295.2 (Loss of 18:2).

**Figure 3 insects-15-00365-f003:**
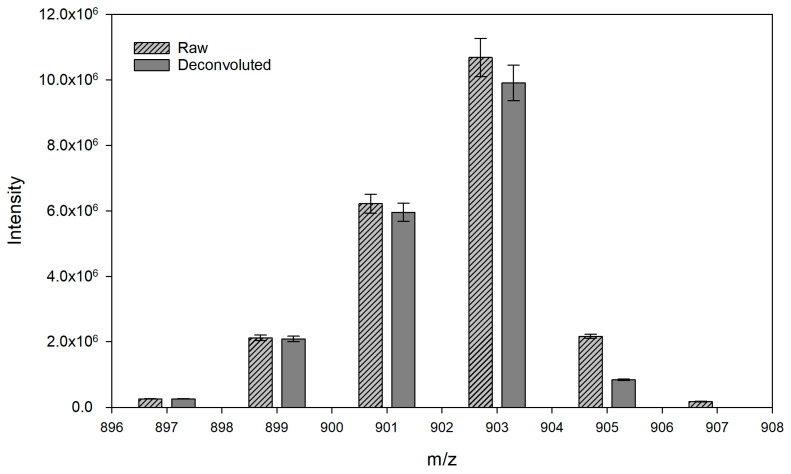
Partial results of isotopic deconvolution in NLS 299.2.

**Figure 4 insects-15-00365-f004:**
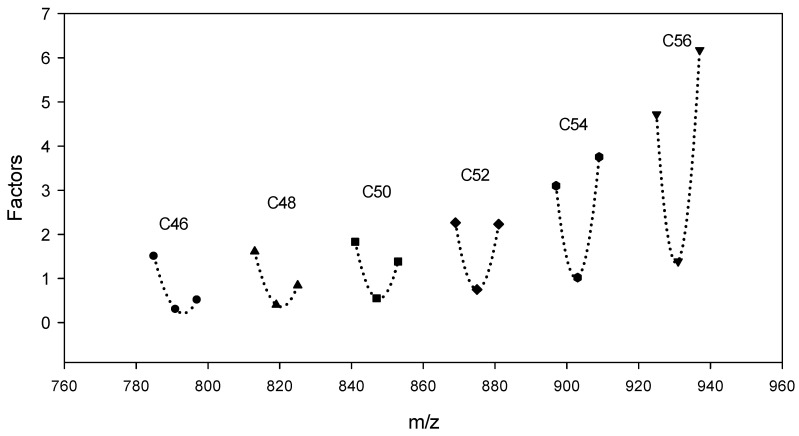
The estimation of the correlation factor along with carbon number using a regression curve for determining the TAG composition in mealworm oil.

**Table 1 insects-15-00365-t001:** The contents of fatty acid in mealworm oil.

Fatty Acid	Chain	Content ^1^ (g/100 g)	RSD (%)
Lauric acid	C12:0	0.42 ± 0.01	1.55
Myristic acid	C14:0	4.07 ± 0.03	0.68
Pentadecanoic acid	C15:0	0.11 ± 0.00	0.55
Palmitic acid	C16:0	17.75 ± 0.09	0.52
Palmitoleic acid	C16:1	1.95 ± 0.01	0.39
Heptadecanoic acid	C17:0	0.1 ± 0.00	0.49
Stearic acid	C18:0	2.49 ± 0.01	0.30
Oleic acid	C18:1	48.68 ± 0.15	0.31
Linoleic acid	C18:2	23.09 ± 0.07	0.29
Arachidic acid	C20:0	0.12 ± 0.00	1.19
Eicosenoic acid	C20:1	0.13 ± 0.00	0.96
Linolenic acid	C18:3	0.96 ± 0.01	0.54
Eicosatrienoic acid	C20:2	0.08 ± 0.01	9.97

^1^ All values are means ± SD of five analyses.

**Table 2 insects-15-00365-t002:** The composition of TAGs in mealworm oil.

m/z	TAG	Composition	Contents		RSD (%)
			nmol/g ^1^	Portion (%)	
793	C46:2	14:0/16:1/16:1	6.6 ± 0.5	0.1	7.1
795	C46:1	14:0/16:0/16:1	5.3 ± 0.5	0.1	9.8
797	C46:0	14:0/16:0/16:0	2.3 ± 0.1	0	4.2
819	C48:3	14:0/16:1/18:2	24.3 ± 0.9	0.3	3.6
819	C48:3	16:1/16:1/16:1	3.4 ± 0.2	0	5.6
821	C48:2	14:0/16:0/18:2	27.4 ± 1.6	0.3	5.7
821	C48:2	14:0/16:1/18:1	62.8 ± 0.7	0.7	1.1
821	C48:2	16:0/16:1/16:1	8.6 ± 0.5	0.1	6.0
823	C48:1	14:0/16:0/18:1	25.4 ± 1.1	0.3	4.2
823	C48:1	16:0/16:0/16:1	3.3 ± 0.1	0	3.4
825	C48:0	16:0/16:0/16:0	2.3 ± 0.2	0	10.8
845	C50:4	14:0/18:2/18:2	75.2 ± 5.7	0.9	7.5
845	C50:4	16:0/16:1/18:3	15.2 ± 0.9	0.2	5.7
847	C50:3	14:0/18:0/18:3	4.2 ± 0.6	0	13.6
847	C50:3	14:0/18:1/18:2	91.7 ± 3.4	1.1	3.7
847	C50:3	16:0/16:0/18:3	4.8 ± 0.4	0.1	8.2
847	C50:3	16:0/16:1/18:2	56.9 ± 5.3	0.7	9.3
849	C50:2	14:0/18:0/18:2	20.4 ± 2.2	0.2	10.9
849	C50:2	14:0/18:1/18:1	105.4 ± 2.9	1.2	2.7
849	C50:2	16:0/16:0/18:2	26 ± 1.5	0.3	5.6
849	C50:2	16:0/16:1/18:1	16.9 ± 1.2	0.2	7.2
851	C50:1	14:0/18:0/18:1	16.2 ± 0.8	0.2	5.0
851	C50:1	16:0/16:0/18:1	56.7 ± 3.3	0.7	5.8
851	C50:1	16:0/16:1/18:0	18.8 ± 0.1	0.2	0.4
853	C50:0	16:0/16:0/18:0	13.8 ± 1.5	0.2	10.7
871	C52:5	16:0/18:2/18:3	167.7 ± 2.5	2	1.5
871	C52:5	16:1/18:2/18:2	156.7 ± 14	1.8	9.0
873	C52:4	16:0/18:1/18:3	105.4 ± 6.8	1.2	6.5
873	C52:4	16:0/18:2/18:2	600.5 ± 56.4	7.1	9.4
873	C52:4	16:1/18:1/18:2	246.2 ± 30.5	2.9	12.4
875	C52:3	16:0/18:0/18:3	7.5 ± 0.7	0.1	8.9
875	C52:3	16:0/18:1/18:2	1488.1 ± 117.7	17.5	7.9
875	C52:3	16:1/18:0/18:2	20.9 ± 1.7	0.2	8.3
875	C52:3	16:1/18:1/18:1	171.7 ± 14.7	2	8.6
877	C52:2	16:0/18:0/18:2	21.5 ± 1.7	0.3	8.1
877	C52:2	16:0/18:1/18:1	1549.4 ± 135.5	18.2	8.7
877	C52:2	16:1/18:0/18:1	17.5 ± 1	0.2	5.9
879	C52:1	16:0/18:0/18:1	91.1 ± 9	1.1	9.9
895	C54:7	18:2/18:2/18:3	136.6 ± 8.9	1.6	6.5
897	C54:6	18:1/18:2/18:3	251 ± 11.9	3	4.7
897	C54:6	18:2/18:2/18:2	383.6 ± 39.7	4.5	10.3
899	C54:5	18:1/18:1/18:3	46 ± 2.4	0.5	5.3
899	C54:6	18:1/18:2/18:2	659.8 ± 27.5	7.8	4.2
901	C54:4	18:0/18:1/18:3	10.3 ± 1	0.1	9.5
901	C54:4	18:0/18:2/18:2	48.4 ± 2.7	0.6	5.7
901	C54:4	18:1/18:1/18:2	870.1 ± 26.3	10.2	3.0
903	C54:3	16:0/18:2/20:1	15.1 ± 0.3	0.2	2.0
903	C54:3	18:0/18:1/18:2	236.1 ± 25.8	2.8	10.9
903	C54:3	18:1/18:1/18:1	328.2 ± 14	3.9	4.3
905	C54:2	16:0/18:1/20:1	8.7 ± 0.3	0.1	3.6
905	C54:2	16:0/18:2/20:0	10.6 ± 0.6	0.1	5.5
905	C54:2	18:0/18:0/18:2	9.4 ± 0.4	0.1	4.1
905	C54:2	18:0/18:1/18:1	42.1 ± 1.4	0.5	3.3
907	C54:1	16:0/18:0/20:1	16.6 ± 1.3	0.2	7.5
929	C56:4	18:2/18:1/20:1	41.1 ± 1.4	0.5	3.5
931	C56:3	18:1/18:1/20:1	26.7 ± 1.8	0.3	6.9
933	C56:2	18:1/18:1/20:0	30.3 ± 0.8	0.4	2.8

^1^ All values are means ± SD of five analyses.

**Table 3 insects-15-00365-t003:** The validation parameters of this method.

Standard	Recovery (%)	Linearity (R^2^)	LOD (μM)	LOQ (μM)
Tripalmitin (16:0/16:0/16:0)	90.9 ± 3.2	0.9993	0.25	0.5
Tristerarin (18:0/18:0/18:0)	86.9 ± 2.4	0.9991	0.2	0.4
Triolein (18:1/18:1/18:1)	95.0 ± 0.9	0.9999	0.05	0.1
Trilinolein (18:2/18:2/18:2)	93.6 ± 4.5	0.9998	0.05	0.1
Trilinoleinin (18:3/18:3/18:3)	94.2 ± 2.8	0.9997	0.05	0.1
1,3-dipalmitoyl-2-oleoylglycerol (16:0/18:1/16:0)	88.1 ± 3.6	0.9990	0.1	0.2
1,3-distearoyl-2-oleoylglycerol (18:0/18:1/18:0)	90.6 ± 1.6	0.9992	0.07	0.14
1-palmitoyl-2-oleoyl-3-stearoyl-sn-glycerol (16:0/18:1/18:0)	92.4 ± 1.9	0.999	0.1	0.2

## Data Availability

The data presented in this study are available on request from corresponding author.

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
