# Peer review of "Determination of Triacylglycerol Composition in Mealworm Oil (*Tenebrio molitor*) via Electrospray Ionization Tandem Mass Spectrometry with Multiple Neutral Loss Scans"

_insects, 2024, doi:10.3390/insects15050365_

Round 1
Reviewer 1 Report
Comments and Suggestions for Authors
In this manuscript an interesting contribution about the TAGs composition of mealworm oil is included. Adequate analytical methods have been applied and sound results have also been obtained. In the following some other comments are included below.
Some more updated references could be included. For instance, ref 19 and 20 could be changed by J. Agric. Food Chem., 2021, 69, 8895-8909, and Anal. Chim. Acta, 2019, 1061, 28-41.
Ref. 35 should be cited in an easier way to be found by interested readers.
Change regressive curve by regression curve along the manuscript
Line 127. Revise thickness of the layer of the stationary phase.
Line 234. Change ...are same concentration. by …have the same concentration.
Line 293. Sentence A detailed study … in Table 2. can be deleted.
Line 340. Revise spelling of Therefore
Line 353. Change quantified by quantitative
Author Response
Summary
First of all, we would like to thank you for your effort to review of our manuscript, titled “Determination of Triacylglycerol in Mealworm Oil (Tenebrio molitor) by Electrospray Ionization Tandem Mass Spectrometry with Multiple Neutral Loss Scan.”
Please find the detailed responses below and the corresponding revisions/corrections highlighted/in track changes in the re-submitted files.
Once again, thank you for your detailed review and It is great opportunity to improve our study. I look forward to positive feedback from you.
Comments 1: In this manuscript an interesting contribution about the TAGs composition of mealworm oil is included. Adequate analytical methods have been applied and sound results have also been obtained. In the following some other comments are included below.
Some more updated references could be included. For instance, ref 19 and 20 could be changed by J. Agric. Food Chem., 2021, 69, 8895-8909, and Anal. Chim. Acta, 2019, 1061, 28-41.
Response 1: Thank you for pointing this out. We agree with this comment. Therefore, we read and checked the recommended references. Line 75, Line 464-447 changed the references.
Comments 2: Ref. 35 should be cited in an easier way to be found by interested readers.
Response 2: Thank you for pointing this out. We agree with this comment. Therefore, we added link to connect Korean food code. Line 506 added link to connect Korean food code.
Comments 3: Change regressive curve by regression curve along the manuscript
Response 3: Line 165, 244, 245, the word changed (regressive -> regression)
Comments 4: Line 127. Revise thickness of the layer of the stationary phase.
Response 4: Thank you for pointing this out, it was our mistake. Therefore, we checked and changed thickness of the layer of the stationary phase. Line 134, the thickness changed (20 -> 0.2)
Comments 5: Line 234. Change ...are same concentration. by …have the same concentration.
Response 5: Line 244, the phrase changed (Change ...are same concentration -> have the same concentration)
Comments 6: -Line 293. Sentence A detailed study … in Table 2. can be deleted.
Response 6: Line 296-297, the sentence was removed
Comments 7: Line 340. Revise spelling of Therefore
Response 7: Line 366, the word corrected
Comments 8: Line 353. Change quantified by quantitative
Response 8: Line 379, the word changed
Reviewer 2 Report
Comments and Suggestions for Authors
In this manuscript, triacylglycerol in mealworm oil has been studied by MS method, which would be useful for insect resource. Thus, this paper would be suitable for this journal after some minor revisions.
Comments:
1) The oxidative stress of triacylglycerol in mealworm oil should be evaluated by some methods. Can more details information be provided?
2) In section "2.2. Analysis of fatty acid", why the pretreatment method for fatty acid analysis use both "sodium hydroxide in a methanol" and "borontrifluoride in methanol"?
Comments on the Quality of English LanguageThe quality of English language is OK.
Author Response
Summary
First of all, we would like to thank you for your effort to review of our manuscript, titled “Determination of Triacylglycerol in Mealworm Oil (Tenebrio molitor) by Electrospray Ionization Tandem Mass Spectrometry with Multiple Neutral Loss Scan.”
Please find the detailed responses below and the corresponding revisions/corrections highlighted/in track changes in the re-submitted files.
Once again, thank you for your detailed review and It is great opportunity to improve our study. I look forward to positive feedback from you.
Comments 1: In this manuscript, triacylglycerol in mealworm oil has been studied by MS method, which would be useful for insect resource. Thus, this paper would be suitable for this journal after some minor revisions.
The oxidative stress of triacylglycerol in mealworm oil should be evaluated by some methods. Can more details information be provided?
Response 1: In first, we agree with your opinion “provide detail information to evaluate oxidative stress of triacylglycerol”. So we checked major references J. Agric. Food Chem. 2005, 53, 5766−5771. In the references, “Oxidative stability was evaluated by an accelerated automated test using the Rancimat apparatus, model CH 9100 (Metrohm Co., Basel, Switzerland). Into Rancimat vessels containing 2.5 g of purified oil were added different amounts of methanolic solution of antioxidants and 0.5 mL of acetone, and then the mixtures were homogenized. The vessels were covered with the heads, placed into the Rancimat apparatus at room temperature, and then heated under an air flow rate of 4 L/h. When the temperature reached 100 °C (∼35 min), the vessels head outlets were connected to the conductivity cells, the air flow rate was increased to 15 L/h, and the measurement time was started. The time taken until there is a sharp increase of conductivity is termed the induction time (IT), and it is expressed in hours. IT was determined by the intersection of the baseline with the tangent to the conductivity curve.”
In our study, we estimated the oxidative stress of triacylglycerol in mealworm oil using their analysis data. So we mentioned our manuscripts the apparatus information. (Line 60-61)
Comments 2: In section "2.2. Analysis of fatty acid", why the pretreatment method for fatty acid analysis use both "sodium hydroxide in a methanol" and "borontrifluoride in methanol"?
Response 2: We would like to explain about your question. The pretreatment for fatty acid analysis have two steps. In the first step, the fatty acids in glyceride were changed to alkaline salts of fatty acid using the reagent methanolic sodium hydroxide. Then in the second step, the alkaline salts of fatty acid were changed to fatty acid methyl ester using reagent borontrifluoride in methanol solution. Especially, the role of borontrifluoride was catalyst in this reaction.
In our manuscript, added more specific explain the pretreatment method for fatty acid analysis. (Line 127-130)
Reviewer 3 Report
Comments and Suggestions for Authors
The paper reports the validation of an analytical method for determining the triacylglycerols in mealworm. The quality of paper is not appropriate for pubblication and some criticism needs to be solved.
The Introduction section needs to better report the hypothesis behind the research. Which is the novelty with respect to the state of art?
In terms of TAGs composition their regioisomers identification is crucial for industry. How authors can valorize the mealworm fat just on estimation of TAGs
Intro lines 62-68: the composition of TAG is crucial for oil/fat purposes, however the TAGs regioisomes play a key role in fats technology. Please consider that aspect.
In order to demonstrate the suitability of the validated method a direct comparison with a conventional method has to be carried out
The validation of method is poor. I suggest to follow the IUPAC methodology. No LOD and LOQ have been determined; the recovery has to be tested in the range of calibration curves using at least two or three different levels of spiking.
The linearity was determined just using tripalmitin, stearin, triolein, trilinolein and trilinolenin; how authors can be sure that the same intensity of ionization is reached on the other TAGS such as C16:0/C18:1/C18:2. The intensity of ionization for each TAG is strictly related to its composition. Again, the position of fatty acids will lead to different ionization.
The discussion of results is poor and it needs to be improved with a comparison with actual literature.
The conclusions have to be reconsidered, a robust validation of the method is required and a comparison with a conventional methodology has to be carried out in order to confirm the suitability of the validated method.
Author Response
Summary
First of all, we would like to thank you for your effort to review of our manuscript, titled “Determination of Triacylglycerol in Mealworm Oil (Tenebrio molitor) by Electrospray Ionization Tandem Mass Spectrometry with Multiple Neutral Loss Scan.”
Please find the detailed responses below and the corresponding revisions/corrections highlighted/in track changes in the re-submitted files.
Once again, thank you for your detailed review and It is great opportunity to improve our study. I look forward to positive feedback from you.
Comments 1: The paper reports the validation of an analytical method for determining the triacylglycerols in mealworm. The quality of paper is not appropriate for pubblication and some criticism needs to be solved.
The Introduction section needs to better report the hypothesis behind the research. Which is the novelty with respect to the state of art?
Response 1: In first, we agree with your opinion “The Introduction section needs to better report the hypothesis behind the research. Which is the novelty with respect to the state of art.” So we checked more references to explain state-of-art analysis method. J. Insects Food Feed 2022, 8, 157-168. In the references, “the author studied about mealworm lipid such as triacylglycerol, phospholipid and fatty acid using HR-MS” so we added this reference to the introduction. Line 80-82, 478-479
Comments 2: In terms of TAGs composition their regioisomers identification is crucial for industry. How authors can valorize the mealworm fat just on estimation of TAGs
Intro lines 62-68: the composition of TAG is crucial for oil/fat purposes, however the TAGs regioisomes play a key role in fats technology. Please consider that aspect.
Response 2: Secondly, we absolutely agree with your opinion “the TAGs regioisomes play a key role in fats technology. Please consider that aspect.” So we checked more references to understand the TAG regioisomeres. Anal. Biochem. 2017, 524, 3-12. In the references, “the author studied about enantiomer and regioisomer of TAGs using appropriate example. So we added this references to the introduction. Line 63-69, 460-461
Comments 3: In order to demonstrate the suitability of the validated method a direct comparison with a conventional method has to be carried out
The validation of method is poor. I suggest to follow the IUPAC methodology. No LOD and LOQ have been determined; the recovery has to be tested in the range of calibration curves using at least two or three different levels of spiking.
The linearity was determined just using tripalmitin, stearin, triolein, trilinolein and trilinolenin; how authors can be sure that the same intensity of ionization is reached on the other TAGS such as C16:0/C18:1/C18:2. The intensity of ionization for each TAG is strictly related to its composition. Again, the position of fatty acids will lead to different ionization.
Response 3: We agree with your opinion “The validation of method is poor.” First of all, we checked your recommendation which is IUPAC validation method. So we added LOD and LOQ data. The new recovery data was added using three different levels of spiking. Also, we agree with your opinion “how authors can be sure that the same intensity of ionization is reached on the other TAGs such as C16:0/C18:1/C18:2. The intensity of ionization for each TAG is strictly related to its composition. Again, the position of fatty acids will lead to different ionization.” Actually, we recognized this aspect so we used adjustment factor calculated using regression curve to explain the differences of TAG ion intensity. We agree with the necessity of actual validation data for other TAGs to confirm our adjustment factor. So, the updated validation was conducted utilized more TAGs standard including 16:0/18:1/16:0, 16:0/18:1/18:0 and 18:0/18:1/18:0. The results of validation were well matched with adjustment factor.
- method and validation -> reference was added (Line 168, 509-510), LOD and LOQ added (Line 169-170), recovery spiking concentration added (Line 172) and the validation of TAGs standard added (Line 175-176)
- Result -> The standard for validation was updated (Line 297-302). New recovery data (Line 303-304), LOD and LOQ data were updated (308-309). All of validation data was updated in the table (Line 311).
Comments 4: The discussion of results is poor and it needs to be improved with a comparison with actual literature.
Response 4: We agree with your opinion “The discussion needs to be improved with a comparison with actual literature.” So we searched more references to compare our data. We added the comparison with actual literature about fatty acid (Line 315-326) and triacylglycerol (Line 353-361), respectively.
Comments 5: The conclusions have to be reconsidered, a robust validation of the method is required and a comparison with a conventional methodology has to be carried out in order to confirm the suitability of the validated method.
.
Response 5: We agree with your opinion “The conclusions have to be reconsidered.” So we added updated validation data (Line 393-395) and mentioned about the limitation of our research (398-402).